# Substantial but spatially heterogeneous progress in male circumcision for HIV prevention in South Africa

Matthew L. Thomas [1,2 ✉], Khangelani Zuma[3,4], Dayanund Loykissoonlal[5], Ziphozonke Bridget Dube[6], Peter Vranken[7], Sarah E. Porter[7], Katharine Kripke [8], Thapelo Seatlhodi[5,9], Gesine Meyer-Rath [10,11], Leigh F. Johnson[9] & Jeffrey W. Imai-Eaton [2,12]

## Abstract

**Background** Voluntary medical male circumcision (VMMC) reduces the risk of male HIV acquisition by 60%. Programmes to provide VMMCs for HIV prevention have been introduced in sub-Saharan African countries with high HIV burden. Traditional circumcision is also a long-standing male coming-of-age ritual, but practices vary considerably across populations. Accurate estimates of circumcision coverage by age, type, and time at subnational levels are required for planning and delivering VMMCs to meet targets and evaluating their impacts on HIV incidence.

**Methods** We developed a Bayesian competing risks time-to-event model to produce region-age-time-type specific probabilities and coverage of male circumcision with probabilistic uncertainty. The model jointly synthesises data from household surveys and health system data on the number of VMMCs conducted. We demonstrated the model using data from five household surveys and VMMC programme data to produce estimates of circumcision coverage for 52 districts in South Africa between 2008 and 2019.

**Results** Nationally, in 2008, 24.1% (95% CI: 23.4–24.8%) of men aged 15–49 were traditionally circumcised and 19.4% (18.9–20.0%) were medically circumcised. Between 2010 and 2019, 4.25 million VMMCs were conducted. Circumcision coverage among men aged 15–49 increased to 64.0% (63.2–64.9%) and medical circumcision coverage to 42% (41.3–43.0%). Circumcision coverage varied widely across districts, ranging from 13.4 to 86.3%. The average age of traditional circumcision ranged between 13 and 19 years, depending on local cultural practices.

**Conclusion** South Africa has made substantial, but heterogeneous, progress towards increasing medical circumcision coverage. Detailed subnational information on coverage and practices can guide programmes to identify unmet need to achieve national and international targets.

## Plain language summary

Voluntary medical male circumcision reduces the risk of male HIV acquisition. Programmes to provide circumcisions for HIV prevention have been introduced in sub-Saharan African countries with high HIV burden. Estimates of circumcision coverage are needed for planning and delivering circumcisions to meet targets and evaluate their impacts on HIV incidence. We developed a model to integrate date from both household surveys and health systems on the number of circumcisions conducted, and applied it to understand how the practices and coverage of circumcision are changing in South Africa. National circumcision coverage increased considerably between 2008 and 2019, however, there remains a substantial subnational variation across districts and age groups. Further progress is needed to reach national and international targets.

[1] Department of Earth and Environmental Sciences, University of Manchester, Manchester, UK. [2] MRC Centre for Global Infectious Disease Analysis, School of Public Health, Imperial College London, London, UK. [3] Human and Social Capabilities Research Division, Human Sciences Research Council, Pretoria, South Africa. [4] School of Public Health, University of the Witwatersrand, Johannesburg, South Africa. [5] National Department of Health, Pretoria, South Africa. [6] Genesis Analytics, Johannesburg, South Africa. [7] Division of Global HIV and Tuberculosis, Centers for Disease Control and Prevention, Pretoria, South Africa. [8] Avenir Health, Washington, DC, USA. [9] Centre for Infectious Disease Epidemiology and Research, University of Cape Town, Cape Town, South Africa. [10] Health Economics and Epidemiology Research Office, Faculty of Health Sciences, University of Witwatersrand, Johannesburg, South Africa. [11] Department of Global Health, Boston University School of Public Health, Boston, MA, USA. [12] Center for Communicable Disease Dynamics, Department of Epidemiology, Harvard T.H. Chan School of Public Health, Boston, MA, USA. ✉email: matthew.l.thomas@manchester.ac.uk

Preventing HIV infections continues to be a major public health priority, particularly in southern and eastern African countries which experience the highest burden. Despite considerable progress combatting the epidemic, new HIV infections remain well above national and international targets[1]. Voluntary medical male circumcision (VMMC), defined as the complete surgical removal of the foreskin, reduces the risk of female-to-male HIV acquisition by 60%[2–6]. Some evidence indicates that VMMC may also reduce the risk of HIV acquisition among men who have sex with men[7] and may decrease the risk of other sexually transmitted infections, such as syphilis, herpes simplex virus type 2, and human papillomavirus[8].

VMMC is appealing as an HIV prevention intervention because it is a one-time, efficient, safe, and cost-effective method. The World Health Organization (WHO) and the Joint United Nations Programme on HIV/AIDS (UNAIDS) identified 15 African countries with high HIV prevalence and low male circumcision (MC) prevalence as priority countries to scale-up VMMC for HIV prevention[9–11]. In 2010, ambitious targets were set to achieve 80% circumcision coverage among men aged 15–49 years by 2015. In 2016, complementary targets were established to reach 90% circumcision coverage among adolescent boys and young men aged 10–29 years by 2021[12]. In South Africa, the South African National Strategic Plan for HIV, TB, and STIs 2017–2022 set a target to provide 2.5 million VMMC over 5 years as part of a comprehensive combination HIV prevention package[13].

While the large-scale provision of medical male circumcision (MMC) for HIV prevention is relatively recent, male circumcision has traditionally been practised in many African countries as part of traditional male initiation ceremonies (TMIC). Circumcisions during TMIC are typically conducted among adolescent boys and young men, but its provision, typical age at circumcision, and what it entails differ considerably between and within countries, influenced by community-established values, and religious, ethnic and cultural identities. In South Africa, there is substantial heterogeneity in traditional circumcision practices across ethnic groups, ranging from rarely conducted to nearly universal[14,15]. Traditional male circumcision (TMC) conducted during TMIC is often performed using non-medical methods in non-clinical settings by a traditional practitioner with no formal medical training[16–18]. Whether TMC involves complete or only partial removal of the foreskin varies by practicing groups. In some cases, TMC may involve partial circumcision or only a simple incision in the prepuce and is not thought to confer the same HIV prevention benefits as medical circumcision[19–21]. Due to incomplete HIV prevention efficacy and safety considerations, VMMC programmes are increasingly working with traditional leaders for MMCs to be conducted as part of TMICs by medical service providers (hereafter referred to as MMC-T for medical male circumcisions conducted in traditional settings). Replacing TMC with MMC-T ensures men circumcised through TMIC have the same protection against HIV infection, as well as safe and sanitary surgical procedures[19].

In light of this heterogeneity, planning and delivering VMMC services to meet programmatic targets requires detailed and timely information about the coverage of male circumcision by age group and type of circumcision (traditional or medical) at subnational levels. Information about male circumcision coverage is available from two sources: (i) household surveys collecting self-reported circumcision status, and (ii) health system programme data on the number of VMMC conducted for HIV prevention. National household surveys, conducted (roughly) every 5 years, collect data on current HIV status and other risk factors including self-reported circumcision status, age at circumcision, and type of circumcision from nationally representative samples. Household survey estimates of circumcision coverage are relatively precise at the national level, but have large statistical uncertainty at subnational levels and within fine age groups due to small sample sizes. Moreover, as household surveys are conducted infrequently, they do not reflect recent intervention implementation and therefore alone do not provide up-to-date information to guide programmes. In South Africa, the numbers of VMMC conducted by HIV prevention services are reported to the National Department of Health (NDoH) by VMMC programme implementers. These data can be used to monitor circumcision coverage and update unmet need targets between surveys. They do not, however, reflect men traditionally circumcised, or those medically circumcised before the start of the national VMMC programme in 2010. Programme implementers conduct annual reviews for programme management and often adjust programme priorities, approach, and funding from year to year. Combining information from both sources of data is therefore important to quantify current circumcision coverage for target setting and strategic decision making.

Previous approaches to estimate district-level circumcision coverage have considered both data sources, but often not together[22–24]. Cork et al. used a Bayesian geostatistical model to produce annual estimates of total male circumcision coverage in sub-Saharan Africa at multiple spatial resolutions between 2000 and 2017[22] using survey data only. Percentage coverage was smoothly interpolated between surveys, but did not incorporate VMMC programme data and therefore unable to account for the rapidly changing landscape in VMMC. The geostatistical prevalence mapping approach focused on predicting male circumcision coverage among only the aggregate age range 15–49 years, and did not track the structured dynamics of circumcision coverage by type within cohorts. The Decision Makers' Program Planning Toolkit, Version 2 (DMPPT2) is a model to support planning VMMC scale-up in sub-Saharan Africa that allows national HIV programme users to enter reported number of VMMCs and generate VMMC targets, coverage estimates, and impact projections[23,24]. DMPPT2 combines estimates of circumcision coverage, derived from survey data prior to the VMMC scale-up, with the reported numbers of VMMCs in a compartmental model to estimate circumcision coverage and unmet need in 5-year age groups at a subnational level over time. The approach is limited as it does not incorporate data from more recent household surveys (since intervention implementation) and does not provide statistical uncertainty associated with the circumcision coverage estimates.

Here, we developed a model to synthesise both survey and VMMC programme data to estimate the probabilities and coverage of medical and traditional male circumcision by subnational area, single-year age, and time with probabilistic uncertainty. The model extends a traditional competing risks time-to-event model in order to integrate both survey data and VMMC programme data, addressing the limitations of previous approaches by Cork et al.[22] and Kripke et al.[23]. We applied the model to estimate annual probabilities and coverage of circumcision by district in South Africa between 2008 and 2019 in order to quantify gaps in attainment of VMMC targets within priority age groups.

## Methods
We created a Bayesian hierarchical model using small area estimation methods for probabilities of circumcision stratified by district, age, and time for two types of circumcision: (1) circumcisions that occurred in traditional male initiation ceremonies or other religious or cultural reasons (TMIC) and (2) circumcisions for non-traditional reasons and/or HIV prevention that take place in a clinical setting using medical methods (MMC-nT).

Reflecting recent efforts to encourage the adoption of medical circumcision methods in TMICs, TMICs were sub-categorised into: (1) traditional circumcisions conducted using non-medical methods (TMC) and (2) circumcisions conducted as part of TMIC but using medical methods (MMC-T), determined by a district- and year-specific probability that TMIC circumcisions were performed as MMC-Ts.

Likelihood functions were specified for the two data sources to inform parameter calibration: (1) the probability of individual-level observations of circumcision age, type, and year reported by men in national household surveys in a time-to-event framework, and (2) the reported number of medical male circumcisions conducted in each district for HIV prevention among males 10 years and older using a Poisson count model. We refer to the following types of circumcision throughout (see Supplementary Fig. 1): MMC-nT: medical male circumcisions conducted outside of traditional male initiation ceremonies, representing the large majority of MMC conducted; TMC: traditional male circumcisions, assumed to be conducted outside a medical setting for traditional male initiation purposes; and MMC-T: medical male circumcisions conducted as part of traditional male initiation ceremonies, typically in place of circumcisions that previously would have been conducted as TMC.

Useful aggregates of these circumcision types are referred to as: MMC: all medical male circumcisions ([MMC-nT] + [MMC-T]), assumed to be consistent with circumcisions reported through VMMC programme data reporting; TMIC: all male circumcisions conducted as part of traditional male initiation practices ([TMC] + [MMC-T]); and MC: all male circumcisions of any type ([MMC-nT] + [TMC] + [MMC-T]).

**Model**. Probabilities of circumcision were modelled using a competing risk discrete time-to-event model[25]. The probability of TMIC was modelled by a logit-linear function with random effects for age, district, and an age-district interaction, allowing for different age patterns of circumcision across districts. We assumed that the probability of TMIC was constant over time as TMIC practices have been relatively stable. The probability of MMC-nT was separated into two processes: (i) paediatric circumcision (for boys aged 0–9 years) and (ii) adolescent and adult circumcision (for those aged 10 years and over). VMMC programmes only provide MMCs for HIV prevention to those aged 10 and over, while infant and paediatric medical circumcision tends to occur through cultural or religious practices unrelated to the scale-up of VMMC for HIV prevention. The probability of paediatric MMC-nT was also modelled using a logit-linear function with random effects for age, district, an age-district interaction, for different patterns of paediatric circumcision across districts. We assumed that the probability of paediatric MMC-nT was constant over time. The probability of adolescent and adult MMC-nT was modelled using logit-linear function with random effects for age, time, and district, and interactions for district-time, age-time, and age-district interactions, to allow for inter-district/time/age variation in circumcision rates. To ensure the probability of any circumcision (i.e. the sum of the probabilities of MMC-nT and TMIC) was bounded by one, we applied the probability of TMIC before MMC-nT in each time step.

The probabilities of MMC-T and TMC were determined by specifying a proportion of TMICs believed to be conducted as MMC-Ts in each year and district. Prior to VMMC programmes, all circumcisions conducted in TMIC were assumed to be TMC. The proportion of TMICs conducted as MMC-Ts may not be identifiable from the survey data, particularly for years since the most recent survey. Thus non-zero proportions of TMICs conducted as MMC-Ts were only specified in districts where

there was knowledge among programme implementers that MMC-Ts were implemented in TMIC settings.

Estimates of circumcision coverage by type within each cohort were calculated using the cumulative incidence function, which defines the marginal probability (or proportion/coverage of) individuals who were circumcised by type (MMC-nT, MMC-T, TMC, TMIC or MMC) by district, age and time, accounting for the competing risk of other circumcision type.

Utilising data on the number of VMMCs reported, we further informed the probabilities of becoming circumcised by district and year. VMMCs conducted by public health programmes consisted of the total number of MMCs (both MMC-nT and MMC-T) conducted. We modelled number of district-age-time MMCs using a Poisson likelihood. To calculate the predicted number of circumcisions conducted per year, we multiplied the male population size with the probability of remaining uncircumcised and the probability of having a MMC-T or MMC-nT.

Prior distributions were specified on all model parameters. We assigned intrinsic conditional autoregressive (ICAR) priors[26] to the district random effects, autoregressive process of order 1 (AR1) priors to the temporal random effects, and penalised B-spline functions on the age random effects. The age-space, age-time, and space-time interaction terms were modelled as Type IV separable interactions as defined by Knorr-Held and Leonard[27], where the precision matrices were constructed using Kronecker products. These priors were specified to pool the district-age-time information on circumcision and allow areas with little to no data to borrow across the stratums as well as capture any correlation in circumcision practices across district, age and time. Diffuse Gaussian priors were assigned to each of the intercepts, and exponential priors were applied to standard deviation parameters. Gaussian priors were specified for all correlation parameters on the logit scale, such that a priori there was a 95% probability that the autocorrelation parameters on AR1 processes were between 0.48 and 0.99 on the real scale. The proportion of TMICs performed as MMC-Ts were assigned independent Gaussian distributions on the logit scale, if there was knowledge of the VMMC programmes to suggest MMC-Ts are taking place. Otherwise, the proportions of TMICs performed as MMC-Ts were fixed at zero.

*Household survey likelihood and accounting for survey weights*. The likelihood function consisted of the product of the likelihoods for two independent data sources: (1) household survey data on individual male respondents' age at circumcision and circumcision type, and (2) VMMC programme data about the number of circumcisions conducted for HIV prevention in each district. Survey data consist of individual observations of self-reported age at circumcision, month of birth, and circumcision type (medical or traditional) for male survey respondents. Each respondent contributed an episode from their year of birth until their year of circumcision (classified as either medical or traditional) or censoring, if they remained uncircumcised at the time of the survey. Observations were right censored if the respondent reported not being circumcised at the time of the survey or left censored if the individual reported being circumcised at the time of survey but did not report their age at circumcision. For circumcised individuals, the year of circumcision was calculated as the year of birth plus the age at circumcision. It was assumed that no circumcisions occurred after age 59, so for uncircumcised individuals, censoring occurred either in the survey year or when they were 59. Individuals who self-reported that they were circumcised but had a missing age at circumcision were included in the analysis through left censoring.

Survey data were collected through a complex two-stage cluster sampling design with unequal sampling probabilities. To account

for these survey designs, the probabilities of circumcision were estimated using a weighted pseudo-likelihood in which we replaced the observed counts with weighted counts, calculated using survey weights. Individuals sampling weight were normalised using the Kish effective sample size[28].

*Assessing model fit.* To assess model fit, we performed posterior predictive checks for the full model (including both survey and VMMC programme data) and the model with only household survey data by constructing posterior predictive distributions survey data observations. We sampled 1000 values for medical or traditional circumcision status for each survey respondent based on predicted circumcision coverage prevalences by district-age-time strata. Both were aggregated to calculate predictive distributions by 5-year age groups (from 0–4 through 55–59).

We compared the mean predictions against the empirical survey estimates using continuous ranked probability scores (CRPS) and error statistics including mean absolute error (MAE) and root mean square error (RMSE). We evaluated the coverage the posterior predictive distributions by calculating the proportion of empirical observations that fell within the 50%, 80%, and 95% quantiles of the posterior predictive distribution.

*Implementation, inference, and prediction.* Models were implemented and fit in R[29] using Template Model Builder (TMB)[30], which uses automatic differentiation and Laplace approximations to estimate posterior distributions for model parameters. Models were optimised using the quasi-newton L-BFGS-B optimisation method[31]. Predictions of the district-age-time probabilities of circumcision and other quantities of interest were estimated using Monte Carlo sampling, drawn from the joint posterior distribution, conditional on the optimised hyper-parameters[32]. Marginal predictive distributions of any quantity of interest, for example, circumcision coverage aggregated to any geographical unit and/or age groups, can then be made by summarising the joint samples generated.

The model estimates full posterior distributions of the annual probabilities of becoming circumcised and the corresponding circumcision coverage in South Africa between 2008 and 2019, by circumcision type, single-year age group, and district. Posterior distributions for aggregates were obtained using Monte Carlo sampling. Samples were drawn from the joint posterior distributions, conditional on the optimised hyper-parameters[32], of the annual probabilities of becoming circumcised and the corresponding circumcision coverage in each region-age-time-type stratum. The joint samples were aggregated into any quantity of interest creating in a samples from the posterior distribution of the quantity of interest. For each aggregate, the posterior mean, median, standard deviation, and quantile-based 95% credible intervals (CI) were computed from the corresponding posterior distribution. Samples were aggregated: (1) from district level to coarser administrative boundaries (province, national), (2) from single-year age groups to 5-year age group (0–4, 5–9, etc.) and coarser priority age groups (15–49, 15–29, etc.) and (3) from individual types of circumcision (MMC-nT, MMC-T and TMIC) to produce combinations including MMC, TMIC and MC. The full list of model outputs is in Supplementary Table 1.

Further technical details on the model used can be found in the Supplementary Methods in the Supplementary Information.

**Data.** South Africa is composed of 52 districts at the second administrative level (admin-2) situated in nine provinces (admin-1). Estimates for the male population size by district and 5-year age group from 2008 through 2019 were sourced from Statistics South Africa Mid-Year Population Estimates 2020[33]. District

population estimates were scaled to align with provincial population estimates by 5-year age group from the Thembisa version 4.4 HIV and demographic model, and distributed to single-year of age according to the proportion of population within each single-year age group from Thembisa v4.4[34].

*Household surveys.* Data on self-reported circumcision status were included from five nationally representative household surveys conducted in South Africa between 2002 and 2017: the South African National HIV Prevalence, HIV Incidence, Behaviour and Communication Survey (SABSSM) from 2002, 2008, 2012 and 2017[35–38] and the South Africa Demographic and Health Survey (DHS) 2016[39]. Information was extracted for 51,261 male respondents across all surveys related to age, district of residence, self-reported circumcision status, age at circumcision, who performed the circumcision and where the circumcision took place, and survey sampling weight. Survey respondents were located to district of residence using survey cluster centroid latitude and longitude.

The specific questions about circumcision status in each survey can be found in Supplementary Table 2. Circumcisions were classified as either medical male circumcisions conducted in non-traditional settings (MMC-nT) or traditional male initiation ceremony circumcisions (TMIC) based on responses to both 'Who performed the circumcision?' and 'Where did the circumcision take place?' using the criteria described in Supplementary Table 3.

*VMMC programme data.* We obtained data on the number of VMMCs performed annually among men aged 10 years and older from South Africa NDoH District Health Information System (DHIS) for South African government fiscal years (April to March) from 2013 through 2019. These were supplemented for the years 2009 through 2012 by district-level data on number of circumcisions recorded in the DMPPT2 model applications[23,24]. For 2018 through 2019, the number of VMMCs conducted in districts in the Eastern Cape province were sourced from data reported to PEPFAR via the Monitoring, Evaluation, and Reporting (MER) Indicator framework instead of NDoH DHIS due to known incomplete reporting of VMMC conducted in traditional male initiation ceremonies (MMC-T) to the NDoH DHIS. The total number of VMMCs performed in each district by data source is shown in Supplementary Figs. 2–5.

*Assumptions about MMC in traditional settings (MMC-T) and migration for circumcision.* Direct data about MMCs conducted in traditional male initiation contexts (MMC-T) were not available. Consequently, strong prior assumptions were required about the proportion of circumcisions conducted in TMIC which were MMC-T, informed by expert knowledge of the national circumcision programme. Within the model, it was assumed that all TMICs were TMC prior to 2013, after which a proportion of TMIC circumcisions conducted were assumed to be MMC-Ts as follows: in Nkangala District in Mpumalanga Province, we assumed that around 90% of the TMICs were MMC-Ts each year from 2013 to 2019. In Waterberg, Capricorn, Vhembe, Sekhukhune, Mopani, Ehlanzeni and Gert Sibande districts (Limpopo and Mpumalanga Provinces), we assumed that an increasing proportion of TMICs were done in a medical context between 2015–2019. Around 20% (in 2015), 40% (in 2016), 60% (in 2017), 80% (in 2018), 90% (in 2019) TMICs are assumed to be MMC-Ts each year. For districts in Eastern Cape province and the City of Cape Town district, we assumed that 99% of men receiving TMICs were MMC-Ts in 2018 and 2019. The majority of men residing in City of Cape Town who received TMIC were Xhosa men who returned to traditional male initiation ceremonies

conducted in Eastern Cape province. In all other districts, we assumed that there were no TMICs done in a medical context.

Many young men who migrate for work return to their family home for traditional male initiation, for example, Xhosa men from the Eastern Cape province to Cape Town and other parts of the Western Cape. Such men appear in household surveys in their district of residence, and the model estimates the probabilities of circumcision among men in their district of residence, not where the circumcision occurred. However, MMC-T provided to these men during TMIC would be recorded in the district where the ceremony occurred. To account for this, the number of VMMCs reported in Eastern Cape province in 2018 and 2019 were re-allocated to all districts in South Africa proportionally to the distribution of men reporting isiXhosa as their primary language in the South Africa 2011 census. The number of re-allocated circumcisions was small in most districts, with most re-allocated to larger metropolitan areas such as Cape Town and Johannesburg (Supplementary Fig. 5).

Further details on the data used can be found in Supplementary Notes 1.

**Ethics statement**. This study involved only secondary analysis of de-identified data and therefore did not require research ethics approval. Survey protocols for the HSRC 2008, 2012, and 2017 surveys were approved by the HSRC Research Ethics Committee (REC: 2/23/10/07; 5/17/11/10; 4/18/11/15, respectively) and the US Center for Disease Control's Institutional

Review Board. For the HSRC 2002 survey, an expert consultation was convened to review legal and ethical issues involved. For the HSRC surveys, all participants provided written or verbal consent. CDC granted a waiver of written consent per 45CFR46 for cases where respondents were unable to provide written consent but consented verbally. Parents and guardians of children under 18 years of age gave informed consent for inclusion of their children in the survey. The South Africa DHS 2016 protocol was reviewed and approved by the SAMRC Ethics Committee and the ICF Institutional Review Board. All respondents provided written informed consent. For never-in-union respondents aged 15–17, both the respondent and parent/guardian provided written consent.

**Reporting summary**. Further information on research design is available in the Nature Portfolio Reporting Summary linked to this article.

## Results

**Model fit**. Figures 1 and 2 illustrate, respectively, the model fit to household survey circumcision coverage data and to VMMC programme data from 2010 through 2019 in each province. Results illustrate how the model reproduces differences in circumcision coverage levels and trends across districts observed in household surveys (Fig. 1a), variation across provinces in traditional versus medical circumcision practices (Fig. 1b), and the impact of VMMC programmes on increasing circumcision

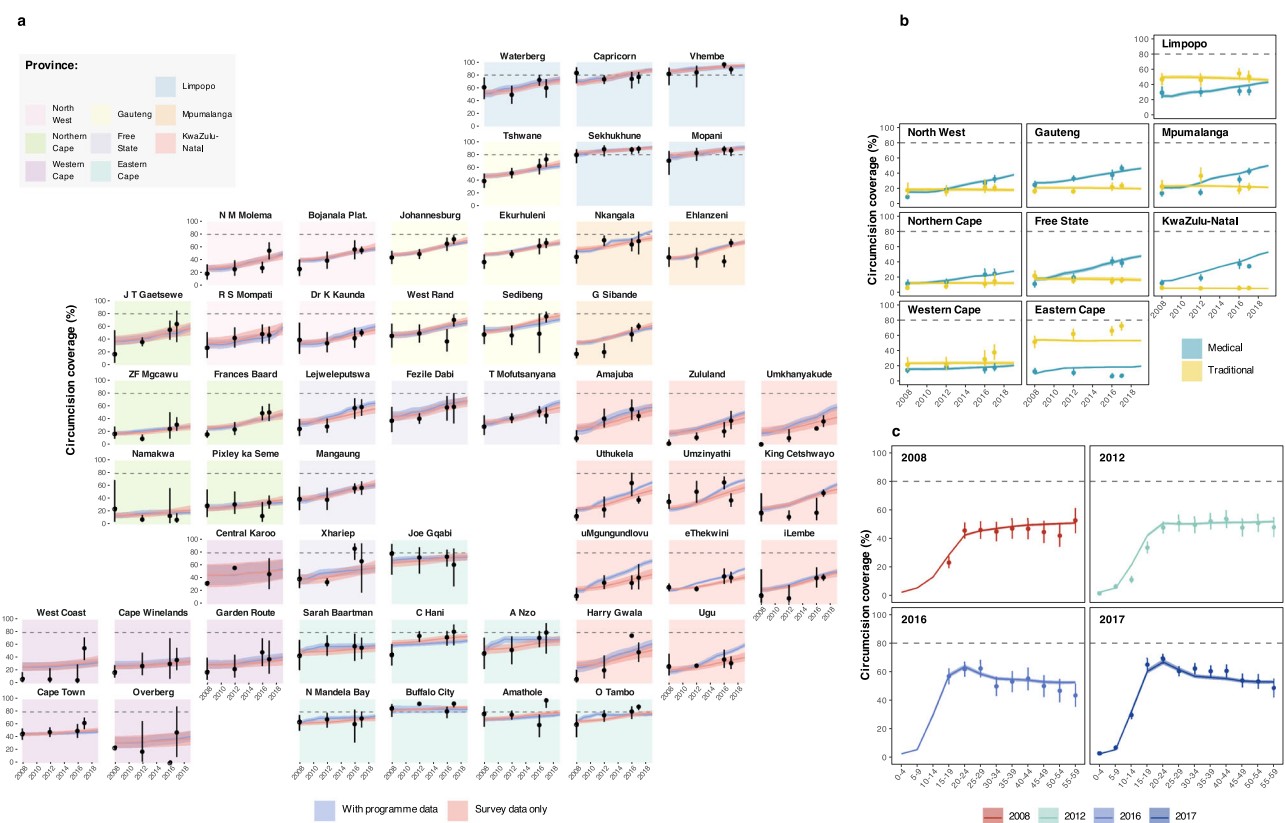

**Fig. 1 Model circumcision coverage estimates compared to direct household survey estimates. a** Estimated male circumcision coverage in men aged 15–49 between 2008 and 2019 by district for models with only household survey data and including survey and VMMC programme data, **b** Estimated medical and traditional male circumcision coverage in men aged 15–49 between 2008 and 2019 by province for the model including both survey and VMMC programme data and **c** estimated national male circumcision coverage in men in 5-year age groups (0–4, 5–9, etc.) in 2008, 2012, 2016 and 2017 for the model including both survey and VMMC programme data. Lines denote the model-estimated mean prevalence, with the shaded regions denoting the 95% CIs. Dots denote the direct survey estimates for circumcision coverage from the 2008, 2012 and 2017 SABSSM surveys and the 2016 DHS survey and the vertical lines denote the associated survey design-based 95% CIs. Dashed horizontal line denotes 80% circumcision coverage target.

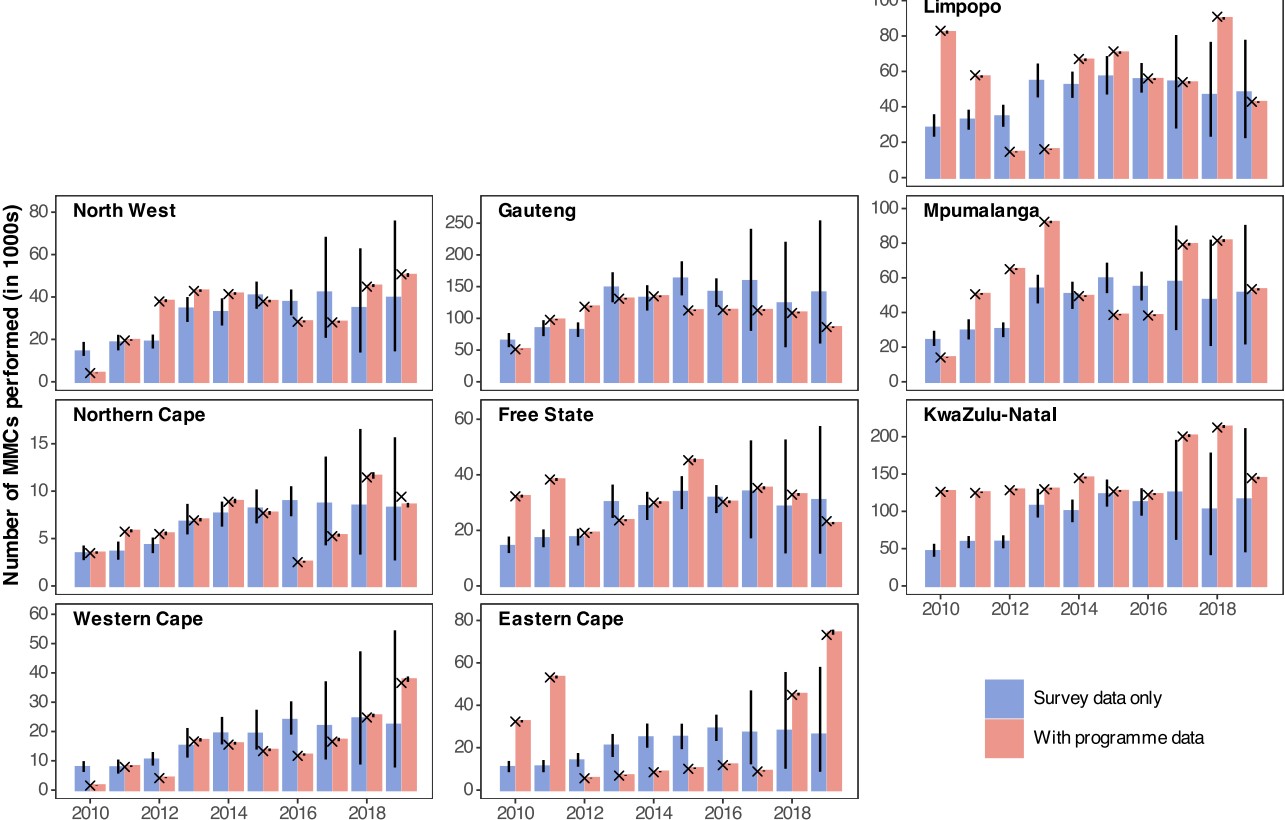

**Fig. 2 Modelled number of medical male circumcisions performed compared to reported VMMC programme data.** Estimated number of medical male circumcisions (MMCs) conducted annually among men and boys over 10 years old between 2008 and 2019 by province for two model versions: with only household survey data (blue bars) and including survey and VMMC programme data in estimation (red bars). Vertical bars denote the model-estimated number of MMCs conducted, with lines denoting the 95% CI. Black crosses denote the reported number of MMC from the VMMC programmes.

coverage particularly among young men aged 15–29 years (Fig. 1c). In the full model including VMMC programme data, the modelled number of VMMCs per year in each province fits closely to reported VMMC programme data in each year (Fig. 2). Model fit to household survey and VMMC programme data at all levels of stratification (national, provincial, and district; aggregated and fine age groups; and circumcision type) are in Supplementary Figs. 6–41. The 50%, 80% and 95% posterior predictive intervals contained 65.4%, 85.3% and 95.0% of medical circumcision coverage observations by district and five-year age group (Supplementary Table 5) and 68.6%, 86.6% and 95.6% of traditional circumcision coverage observations (Supplementary Table 6). Full model fit summaries by survey and age group can be seen in Supplementary Tables 4–6.

**National circumcision coverage**. Nationally, between 350,000 and 650,000 VMMCs were conducted in South Africa each year from 2010 to 2019 (Fig. 3). Circumcision coverage among men aged 15–49 years was 64.0% (95% CI: 63.2–64.9%) in 2019, an increase of 20.5% (20.0–21.0%) since 2008. Traditional circumcision was more common in 2008 with a TMC coverage of 24.1% (23.4–24.8%) among men 15–49 years compared with MMC coverage of 19.4% (18.9–20.0%). This reversed by 2019 when MMC coverage was 42.0% (41.3–3.0%) compared with a decrease in TMC coverage to 22.0% (21.3–22.7%).

Circumcision coverage increased most among men aged 10–24 years (Fig. 4), priority ages targeted by the South African VMMC programme. Coverage peaked at 74.2% (72.3–76.5%) in men aged 21 years in 2019, an increase of 32.2% (30.3–34.4%) since 2008.

The large change was attributable due to increases in medical circumcision. The highest MMC coverage in 2019 was 61.3% (59.2–63.6%) in men aged 19 years, an increase of 44.9% (42.6–47.3%) between 2008 and 2019. Conversely, TMC coverage among men aged 10–24 years decreased. In 2019, TMC coverage in men aged 19 was 11.0% (10.6–11.5%), a decrease of 9.4% (9.0–9.9%) since 2008.

**Subnational circumcision coverage**. Coverage varied considerably across districts of South Africa (Fig. 5). In 2008, prior to the scale-up of VMMC, male circumcision coverage was highest in Vhembe, Buffalo City, and Sekhukhune districts among men aged 15–49 years with 86.3% (81.4–89.8%), 84.6% (77.6–90.7%), and 80.6% (76.8–83.6%) coverage, respectively. Namakwa, Umkhanyakude, and Zululand districts had the lowest coverage, at 13.4% (9.4–18.9%), 16.9% (11.7–21.7%), and 17.4% (12.1–24.1%). This large range in the total circumcision coverage was due to considerably different traditional circumcision practices across South Africa. Districts in Eastern Cape, Limpopo, and Mpumalanga provinces had high levels of traditional circumcision, reaching over 90% coverage among men by age 30 years in Buffalo City district (Fig. 6). These districts have large populations of Ndebele, Xhosa, Pedi, Venda, and Tsonga people, who all typically perform TMC as part of ritual transition to male adulthood. The lowest levels of traditional circumcision were in KwaZulu-Natal, where less than 10% of men were traditionally circumcised. In 2008, before VMMC programme scale-up medical circumcision coverage was relatively low in most districts, ranging between 3.5% (1.6–5.5%) among men aged 15–49 years

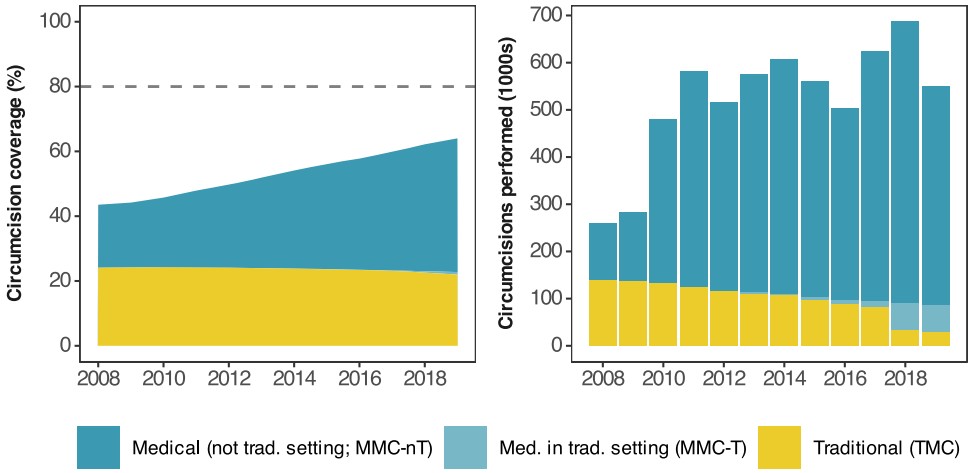

**Fig. 3 National circumcision coverage among men 15–49 years and annual circumcisions performed by type, 2008–2019.** National male circumcision (MC) coverage among men aged 15–49 years between 2008 and 2019, stratified by circumcision type and the estimated number of circumcisions performed annually between 2010 and 2019 stratified by type. Shaded areas represent the posterior mean. Horizontal dashed line denotes the target circumcision coverage of 80%. Bars represent the posterior mean.

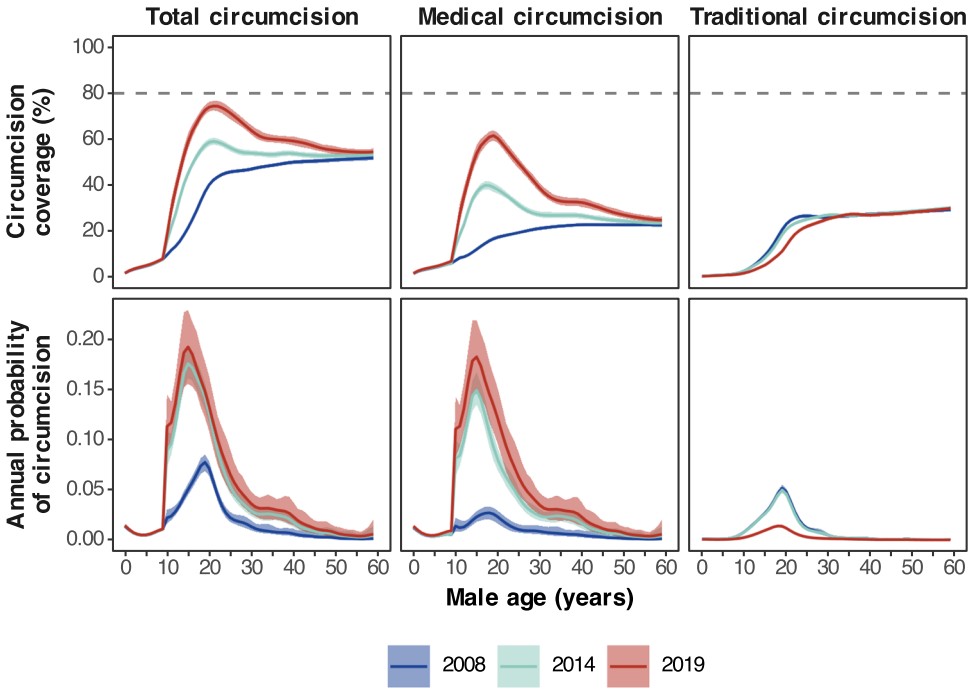

**Fig. 4 Circumcision coverage and annual circumcision probability by age in 2008, 2014, and 2019.** National total (MC), medical (MMC) and traditional (TMC) coverage and probabilities by age in 2008, 2014 and 2019. Lines denote the posterior mean with shaded regions denoting the 95% CI. Dashed line denotes the target circumcision coverage of 80%.

in Chris Hani District and 34.7% (29.7–41.2%) in Capricorn District.

Circumcision coverage increased in all districts in South Africa since the implementation of VMMC national campaigns in 2010 (Fig. 7). Increases have not been uniform across districts as VMMCs have typically been prioritised in areas where HIV incidence and prevalence are high and where there is higher acceptance of VMMC as HIV prevention[11]. Over half of the VMMCs performed were conducted in the three provinces with the highest HIV prevalence among 15–49-year olds: KwaZulu-Natal, Mpumalanga, and Free State[40]. Consequently, the districts with the largest changes in circumcision coverage between 2008

and 2019 were also located in these provinces, with increases of 47.2% (44.9–49.6%), 46.0% (43.4–48.1%) and 42.9% (40.4–45.2%) in uMgungundlovu, Umzinyathi, and Uthukela districts, respectively. In 2019, Vhembe, Mopani, and Sekhukhune districts had the highest circumcision coverage among men aged 15–49 years with 94.2% (90.1–96.7%), 91.6% (88.0–94.6%) and 91.6% (88.9–93.9%) coverage. Namakwa, ZF Mgcawu, and West Coast district had the lowest coverage in 2019 at 18.7% (15.2–23.3%), 28.0% (25.3–31.0%) and 32.9% (27.6–39.6%) coverage. Medical circumcision coverage among men aged 15–49 years ranged considerably in 2019 from 11.6% (10.1–13.0%) in Chris Hani to 64.5% (62.2–66.9%) in Umzinyathi. TMC coverage decreased

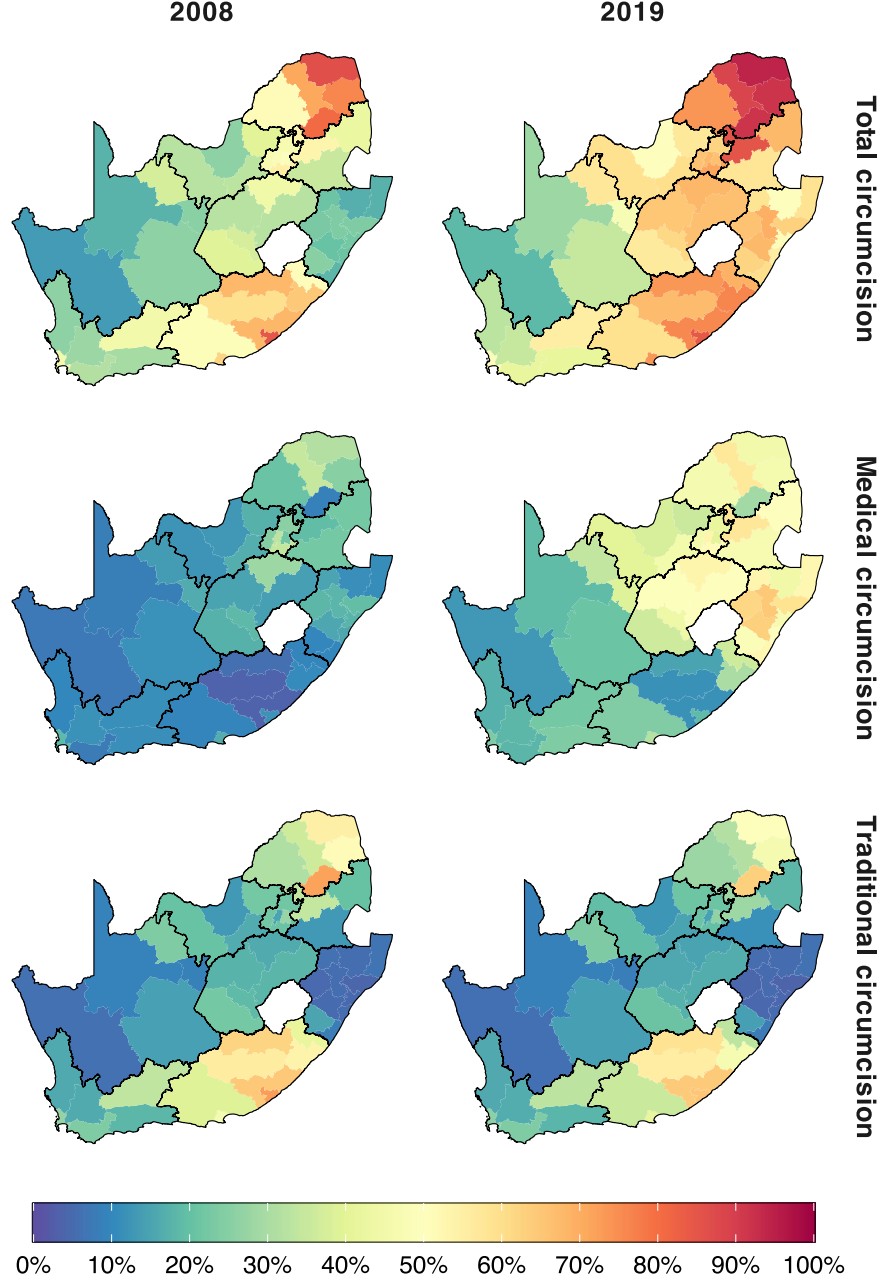

**2008**          **2019**

*Total circumcision*

*Medical circumcision*

*Traditional circumcision*

0%  10%  20%  30%  40%  50%  60%  70%  80%  90%  100%

**Fig. 5 Circumcision coverage by district among men 15–49 years, 2008 and 2019.** Estimated total (MC), medical (MMC) and traditional (TMC) circumcision coverage for men aged 15–49 in each district in 2008 and 2019. Colours denote the posterior mean.

between 2008 and 2019, with the largest decreases in Sekhukhune, Alfred Nzo, and Oliver Tambo of 8.6% (7.6–9.6%), 8.0% (6.2–10.0%), and 7.6% (6.4–8.7%), respectively.

**Subnational variation in practice of and age at circumcision.** The distribution of age at circumcision varied by circumcision type and geography (Fig. 8). Nationally, for circumcisions conducted in 2018, the average age of medical circumcision was 18.4 (95% CI: 17.4–19.5) years and for traditional circumcisions was 17.4 (17.2–17.8) years. The average age of traditional circumcision was the lowest in Limpopo province for both traditional (12.2 years; 11.9–12.8) and medical circumcision (14.7 years; 13.8–15.8). Mpumalanga province also had a lower average of traditional circumcision (15.2 years; 14.6–16.1) than the national average. In other provinces, traditional circumcision typically

occurred around age 18. Western Cape also had a substantially lower average age of medical circumcision (15.6 years; 13.7–17.4) due to the considerable number of circumcisions at birth, which are not related to VMMC programme circumcisions.

**Progress to national and international targets.** By 2019, only 6 districts (Vhembe, Mopani, Sekhukhune, Capricorn, Nkangala and Buffalo City) out of 52 and 1 province (Limpopo) out of 9 were estimated to have achieved the 80% circumcision coverage target in men aged 15–49 years (Table 1 and Supplementary Data 1) and no districts reached the ambitious targets of 90% coverage in adolescent boys and young men aged 10–29 years. However, twelve districts achieved 80% coverage in the 15–24 years age group (Mopani, Vhembe, Nkangala, Capricorn, Sekhukhune, Umzinyathi, uMgungundlovu, Uthukela, Ehlanzeni,

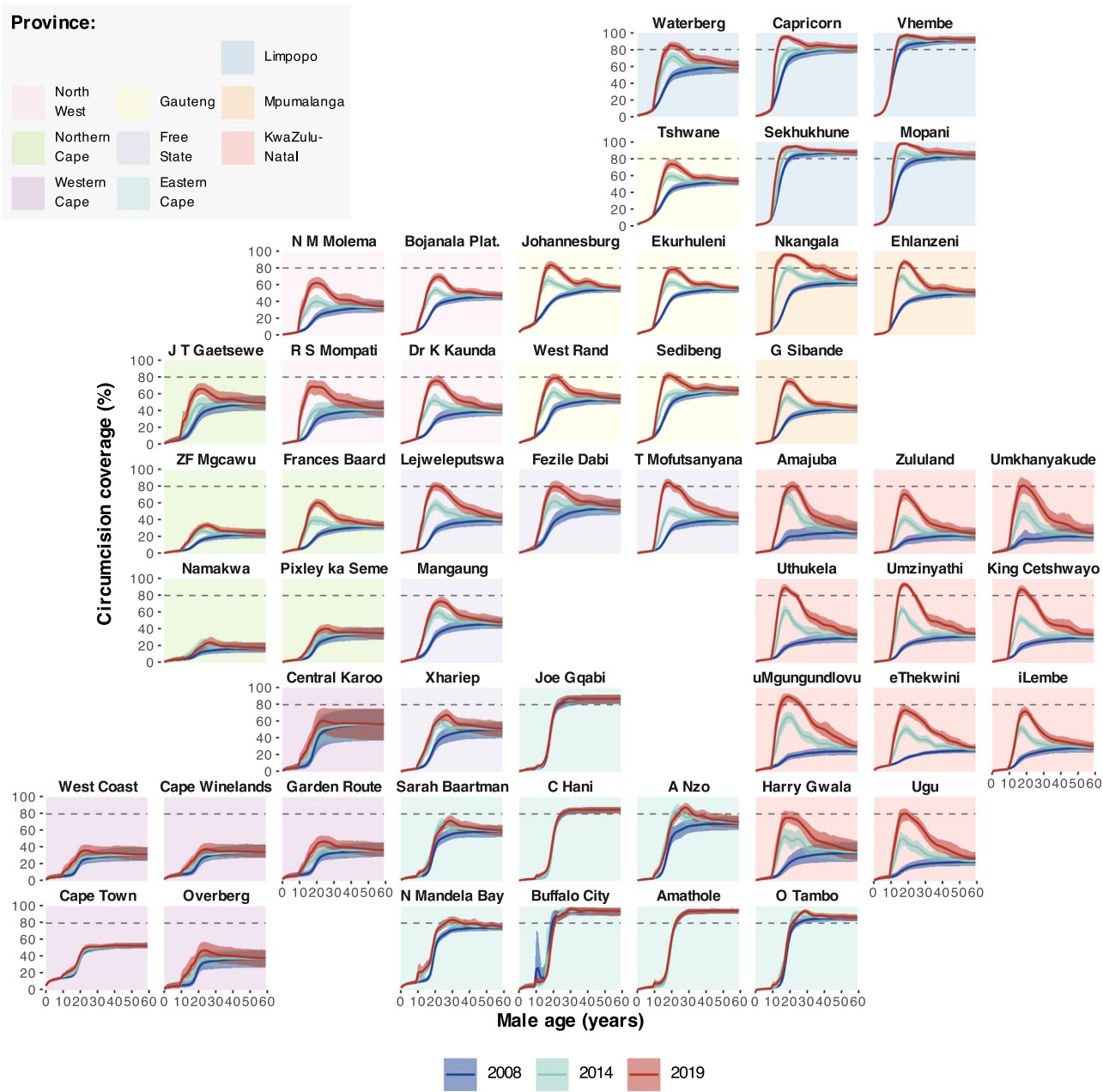

**Fig. 6 Total circumcision coverage by age and district in 2008, 2014 and 2019.** District-level total circumcision (MC) coverage by age in 2008, 2014 and 2019. Lines denote the posterior mean with shaded regions denoting the 95% CI. Horizontal dashed line denotes the target circumcision coverage of 80%. Pastel panel background shading indicates the province within which each district is situated.

King Cetshwayo, Waterberg, T Mofutsanyana, Johannesburg and Sedibeng). Between 2017 and 2019, over 1.7 million VMMCs were delivered. Meeting the South African Government's 2.5 million VMMCs target would require a further 800,000 VMMCs by the end of 2022. In 2019, there were estimated to be 5.43 million (4.83–5.67 million) uncircumcised men aged 15–49 years, with the largest number located in metropolitan areas of Cape Town (623,000; 95% CI: 562,000–659,000), Johannesburg (554,000; 451,000–614,000), and eThekwini (480,000; 360,000–528,000).

**Effect of jointly synthesising multiple data sources.** The primary model results utilised both household survey and VMMC programme data to estimate male circumcision coverage. Models

with and without the programme data included produced similar coverage estimates in most provinces, but estimates were much more precise in the version including programme data (Fig. 1). Nationally, when excluding programme data, circumcision coverage in 2019 was 62.4% (59.3–66.5%), 1.6% lower and with a four-fold wider uncertainty range than in the primary results with program data included in the model (64.0%; 63.2–64.9%). This overall correspondence indicated a high level of agreement in the estimated number of MMCs conducted from the survey data with the number of MMCs conducted by VMMC program providers in South Africa (Fig. 2). Exceptions to this similarity included some districts such as Ugu where coverage was substantially higher by 2019 when including programme data (60.1%; 95% CI: 57.0–64.1% among men 15–49 years) than excluding programme data (41.0%; 39.1–49.2%; Fig. 1a). This is due to higher numbers

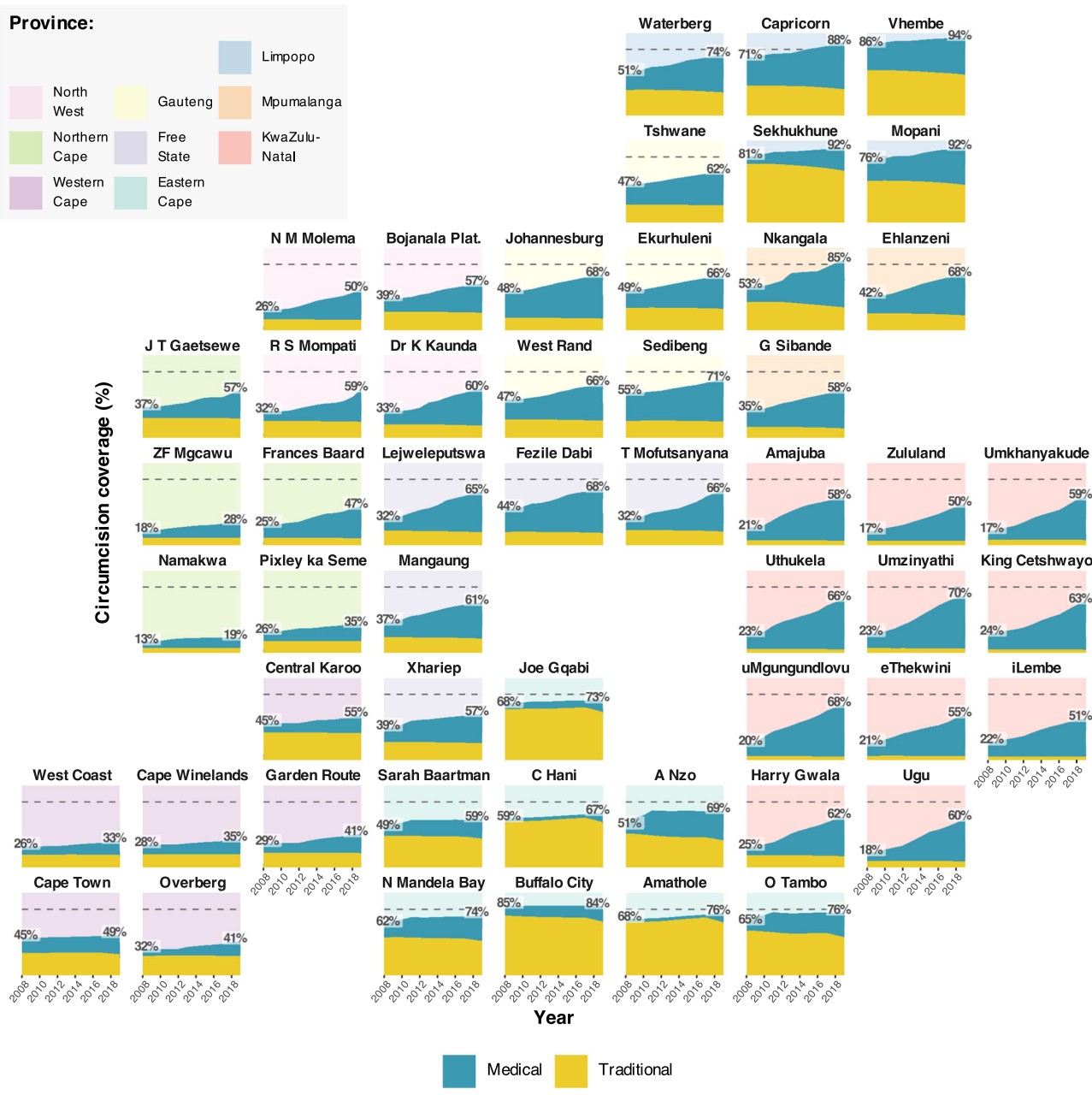

**Fig. 7 Circumcision coverage among men aged 15–49 by district, 2008 to 2019, stratified by circumcision type.** District-level male circumcision coverage among men aged 15–49 years between 2008 and 2019, stratified by circumcision type. Shaded areas denote the posterior mean. Horizontal dashed lines denote the 80% circumcision coverage target. Plots are organised by geography. Pastel panel background shading indicates the province within which each district is situated.

of VMMCs reported in Eastern Cape and KwaZulu-Natal than would be anticipated by using survey only, particularly in the early years of the programme (Fig. 2a).

Posterior predictive comparisons showed similar goodness-of-fit to observed household survey for both the full model (including both survey and VMMC programme data) and the model with only household survey data (Supplementary Tables 4–6). Overall, the coverage of posterior predictive point estimates was similar to that of the survey data at the national, provincial and district level when split by age, time and type (Supplementary Tables 4–6). The posterior predictive checks showed that 66.8% and 96.5% of the medical circumcision coverage estimates fell within the 50% and 95% CIs of the posterior distribution for the survey-only model (Supplementary

Table 4). These decreased slightly to 65.4% and 95.0% for the full model, attributable to the addition of the programme data increasing the precision of the estimates. Posterior predictive interval coverage for traditional circumcision survey observations estimates were similar with 68.6% and 96.0% falling within the 50% and 95% CIs of the posterior distribution for the survey-only model (Supplementary Table 5). Results were very similar when including VMMC programme data with the equivalent coverage being 68.6% and 95.6%, respectively.

## Discussion
Estimates of male circumcision coverage by age and type of circumcision (traditional or medical) over time at subnational levels

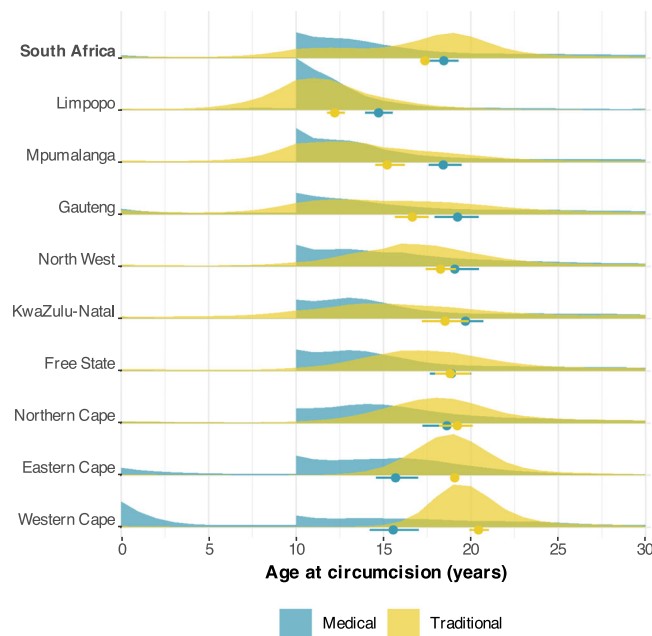

**Fig. 8 Distribution of age at circumcision for medical and traditional circumcisions in 2019.** Results are shown for South Africa overall and for each province. Dots below each density denote the average age of circumcision, with the bars denoting the 95% CI for the mean age of circumcision.

**Table 1 National and provincial summary of circumcision coverage and unmet need in 2019.**

| Region | Coverage | | | Change since 2008 | Population (in 1000s) | |
| --- | --- | --- | --- | --- | --- | --- |
| | Total | Medical | Traditional | Total | Circumcised | Uncircumcised |
| South Africa | 65.7% (64.2–69.5%) | 44.0% (42.4–47.8%) | 21.6% (20.9–22.3%) | 22.2% (20.8–26.0%) | 10,398 (10,165–11,003) | 5432 (4827–5666) |
| Eastern Cape | 74.2% (72.5–78.3%) | 25.3% (23.3–29.5%) | 48.9% (47.6–50.2%) | 9.9% (8.3–13.4%) | 1133 (1108–1196) | 395 (332–420) |
| Free State | 65.9% (63.2–69.6%) | 49.9% (47.3–54.4%) | 16.0% (14.2–18.7%) | 29.8% (27.7–34.3%) | 487 (467–514) | 252 (225–272) |
| Gauteng | 67.7% (65.5–72.3%) | 48.3% (46.0–53.2%) | 19.3% (17.9–21.3%) | 19.4% (17.9–24.0%) | 3202 (3301–3423) | 1530 (1309–1632) |
| KwaZulu-Natal | 61.5% (58.8–68.3%) | 56.1% (53.4–62.8%) | 5.3% (4.6–6.0%) | 40.5% (38.2–46.8%) | 1810 (1731–2011) | 1135 (935–1215) |
| Limpopo | 89.6% (88.0–91.3%) | 46.2% (44.4–48.8%) | 43.4% (41.5–45.5%) | 15.0% (14.0–16.5%) | 1199 (1178–1222) | 139 (116–160) |
| Mpumalanga | 73.1% (71.0–77.2%) | 53.8% (51.9–57.5%) | 19.3% (18.0–20.8%) | 28.9% (27.2–33.1%) | 929 (902–980) | 341 (290–368) |
| Northern Cape | 42.3% (39.0–47.5%) | 30.2% (27.7–34.9%) | 12.1% (10.2–14.3%) | 17.7% (15.3–22.9%) | 120 (111–135) | 164 (149–173) |
| North West | 58.2% (55.6–61.3%) | 40.4% (38.1–44.0%) | 17.8% (15.7–19.8%) | 23.9% (22.0–27.2%) | 651 (622–686) | 468 (434–497) |
| Western Cape | 46.2% (43.8–50.9%) | 23.3% (20.9–26.3%) | 23.0% (20.9–25.2%) | 6.5% (5.2–9.5%) | 866 (821–954) | 1008 (920–1054) |

Estimated total, medical (MMC) and traditional (TMC) circumcision coverage for 2019 among men aged 15–49, along with the absolute change in total circumcision coverage from 2008 to 2019 and the number of circumcised and uncircumcised men aged 15–49 in 2019.

are essential for planning and delivering VMMC programmes. These data support attaining HIV prevention programme targets and evaluating the impact of VMMC campaigns on HIV incidence. In this paper, we developed a model to produce region-age-time-type specific probabilities and coverage of male circumcision, along with associated measures of uncertainty. The model extends a competing risks time-to-event model to integrate both survey data and VMMC programme data, building on previous approaches from Kripke et al.[24] and Cork et al.[22], which did not model both data sources formally.

Our results highlight considerable heterogeneity in circumcision coverage across the 52 districts of South Africa and in the changes between 2008 and 2019. Circumcision coverage increased in all districts in South Africa. The largest increases were in KwaZulu-Natal province, which has the highest HIV prevalence in South Africa and lowest practice of traditional circumcision. Traditional circumcision coverage decreased over the period due to the replacement of traditional circumcisions with medical circumcisions conducted in traditional male initiation settings and men circumcised through VMMC interventions before or instead of through TMIC. While circumcision has increased dramatically, over 5 million men aged 15–49 years remained uncircumcised and there were substantial gaps to reaching 80% coverage among men aged 15–49 years in many districts. However, 12 districts achieved greater than 80% coverage of medical circumcision in the 15–24 years age group among whom intervention circumcisions focused. Continuing to provide VMMC services on this scale and focusing programmes in specific areas will ensure more districts will achieve these targets in the near future.

The granular results can be used to identify locations and key age groups in which either circumcision coverage is low or there are large numbers of uncircumcised men. Stratifying by circumcision type enables VMMC programmes to make programmatic decisions about how to intervene in areas where traditional

circumcision is common. For HIV prevention purposes, it may be beneficial to offer "re-circumcision" if TMCs are only partial circumcisions that do not involve complete removal of the fore-skin. Furthermore, when assessing the impact of the VMMC programmes on HIV incidence, our estimates stratified by circumcision type enable models to account for different protective effects of full medical circumcision versus traditional circumcision.

This model can be applied by circumcision programmes in other countries with household surveys and VMMC programmes for HIV prevention. The estimated numbers of MMCs conducted were largely similar in models with and without the inclusion of programme data; however, the difference was notable in KwaZulu-Natal. It is therefore important to consider and reconcile both data sources when estimating circumcision coverage. Interpreting VMMC programme data required detailed knowledge of the chronology of the South Africa VMMC programme and incorporating them into the model involved specific model features to account for local nuances, such as when integration of medical circumcision into traditional male initiation ceremonies occurred, migration for circumcision, and changing policies around circumcision age. This bespoke data review and model development process for VMMC programme data will likely be similarly time intensive for applications to other settings. However, as a first step, we recommend applying the model with survey data only to furnish district/age/type stratified estimates, which can be triangulated with VMMC programme data and other local information.

The typical age at circumcision varied across provinces for both traditional and medical circumcisions. In Limpopo and Mpumalanga provinces, where the majority of the circumcising population were Ndebele, Pedi, Venda, and Tsonga, circumcision was traditionally between age 12 and 15 years old. In Eastern Cape, Western Cape, Free State, and North West, most of the circumcised population were Xhosa, Sotho, and Tswana, and typical age at circumcision was 18–19 years. Since 2016, the U.S. President's Emergency Plan for AIDS Relief (PEPFAR), the major funder of VMMC programmes in South Africa and other countries in the region, has prioritised circumcising men age 15–29 due to the delayed epidemiological impact of circumcising boys below age 15[10]. Since 2020, PEPFAR has discontinued reimbursement for MMCs among boys below the age 15 years following some evidence of an increased rate of medical complications among this age group[41]. Through domestic co-funding arrangements, South Africa has continued providing VMMC to boys aged 10–14 years to reach them coinciding with preferred traditional circumcision ages. Other countries may also need to reconsider circumcision age restrictions, and locally tailor target age ranges, to reach men with safe and effective VMMC that aligns with local practices and preferences.

Several circumcision dynamics are anecdotally known, but have limited data to quantify and required strong modelling assumptions. The first was the proportion of traditional circumcisions conducted using medical methods (MMC-T) in recent years. Ensuring safe and complete circumcisions for young men has been a major initiative for the VMMC programme. We made prior assumptions about this proportion based on expert knowledge of programme managers and evidence of large number of MMCs reported in districts that have high levels of traditional circumcision. In 2020, the national health information system began explicitly recording the number of MMC-T, so quantitative data on this will be available for future analyses. The second uncertainty was about re-circumcision of men who were previously traditionally circumcised. We assumed that rates of re-circumcision were sufficiently low that it was not important to incorporate into the model. Future household surveys in South

Africa plan to separately capture data on medical and traditional circumcision, which will substantiate or guide revision of this assumption. However, re-circumcision of those who previously underwent only partial circumcision could be important for maximising the HIV prevention impact of VMMC.

Estimates presented here end in 2019 due to data availability for this analysis. The COVID-19 pandemic created substantial short-term disruption to VMMC provision in South Africa[42] and temporary suspension of traditional male initiation ceremonies in several provinces[43]. Nationally, the reported number of VMMCs conducted declined from 464,000 in 2019 to 144,000 in 2020 and 388,000 in 2021. District-level data on changes in numbers of VMMCs since 2019 could be directly incorporated into an extension of this model, but, to capture the full dynamics of the circumcision programme during the COVID-19 period, the model will also need to be updated to replace the assumption of stable traditional circumcision practices with time-location specific information about suspension of traditional male initiation ceremonies during COVID-19.

A further national household survey, SABSSM VI, is currently underway in South Africa, with full results anticipated in late 2023. This survey will provide further data, which can be incorporated into this model, about the scale-up of VMMC during the 2017–2022 National Strategic Plan and how both medical and traditional circumcision practices changed among young men over the COVID-19 period. This survey will also collect information about the extent of the replacement of traditional circumcision by medical circumcision and VMMC re-circumcision of men previously partially circumcised through TMIC.

The models presented here have several limitations and opportunities for further development. First, surveys typically record self-reported circumcision status, which may be susceptible to misreporting as a result of social desirability bias. Many cultures that promote male circumcision as a rite of passage into manhood and some studies using physical examinations have shown that there is some misreporting in circumcision status[44,45]. Second, the classification of circumcision type is based on self-reported questions from surveys, which may be subject to similar misreporting errors due to confusion between medical and traditional circumcision practices. Third, a complete-case analysis was used, that did not adjust for non-response bias in the self-reported circumcision status. Fourth, we relied on a self-reported age at circumcision, which may be subject to recall bias and age-heaping. Fifth, the models used did not make use of any covariates that are predictive of circumcision coverage, such as ethnicity, culture, or religion, to improve the precision of the estimates. Extending the models in this way could allow for further targeting of VMMC services for specific population groups. Sixth, the model did not account for any uncertainty about the male population sizes by age and district. Small area population estimates are uncertain in many countries where the most recent census was long ago (in South Africa, the last census was in 2011, with results of the 2022 census expected in mid-2023). Fifth, similar to other Bayesian model-based analyses of household survey data, we used a weighted pseudo-likelihood approach to approximately account for the complex survey design. Best approaches to account for survey design in model-based analyses remain an active statistical research area[46,47]. Finally, the model did not explicitly represent migration between districts and its impact on district probabilities of circumcision and the corresponding coverage over time. It is suspected that many young men in South Africa move away from rural homes for work and return for their traditional male initiation ceremony. This is particularly the case with Xhosa men from Eastern Cape who move to Western Cape for work. When compiling the programme dataset used in the case study in this paper, we

redistributed MMC-T conducted in the Eastern Cape to Xhosa populations residing in the Western Cape and Gauteng under the assumption that this is the normal place of residence for some men circumcised in the Eastern Cape, but future work should more explicitly model these migration dynamics and how they have changed over age and time.

South Africa has made considerable progress over the past decade towards increasing coverage of medical male circumcision for HIV prevention, with the largest increases among districts in KwaZulu-Natal province, which have the lowest coverage of traditional circumcision and the highest HIV burden. However, progress has been unequal across the country, and, as of the end of 2019, over 5.4 million adult men aged 15–49 years remained uncircumcised. Estimating rates of both medical and traditional circumcision by age over time elucidated large heterogeneity across districts in both practice of traditional circumcision and the age at which it occurs. This underscores the need for VMMC programmes to be tailored at the subnational level to effectively reach young men in line with local norms and practices. The analytical methods proposed here for combining analysis of multiple household surveys with up-to-date VMMC programme data provide a tool which programmes can use to identify and target unmet needs more precisely as margins start to diminish in certain regions. Future household survey data will allow us to address key uncertainties about how circumcision practices have changed as VMMC programmes have scaled-up, including (i) the extent to which traditional circumcision has been replaced by medical circumcision, (ii) levels of re-circumcision for HIV prevention in men who have been previously traditionally circumcised, and (iii) how both traditional and medical circumcision practices changed during the COVID-19 pandemic.

## Data availability

Household survey data are available by request to the South Africa Human Science Research Council (HSRC; http://datacuration.hsrc.ac.za/content/view/access-to-data) and The Demographic and Health Survey (DHS) Program (https://dhsprogram.com/methodology/survey/survey-display-390.cfm). Information on the number of VMMCs performed was obtained from DMPPT2 and South Africa National Department of Health (NDoH). DHIS are by data request to the South Africa NDoH. Data on VMMCs conducted from PEPFAR MER are available from https://data.pepfar.gov/datasets#PDD. Estimates of population were obtained from Thembisa and Statistics South Africa, which are available at https://thembisa.org/downloads and http://www.statssa.gov.za/?page_id=1854&PPN=P0302&SCH=72634, respectively. Estimates of the proportions of men by primary language group in South Africa were obtained from the South Africa 2011 census, which is available from https://www.datafirst.uct.ac.za/dataportal/index.php/catalog/485. The source data underpinning the results presented in this manuscript can be found in Supplementary Data 2–9.

## Code availability

The code used to implement the models and produce the results presented in this manuscript is available from https://github.com/mrc-ide/zaf-circumcision-paper. We have also archived the code on Zenodo and this is available from https://doi.org/10.5281/zenodo.8421386. An R package implementing an updated and extended version of the methods from this manuscript is currently being constructed and is available from https://github.com/mrc-ide/threemc.

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

## Acknowledgements

This research was supported by the American people and the President's Emergency Plan for AIDS Relief (PEPFAR) through USAID under the terms of Cooperative Agreement 72067419CA00004 to HE2RO and through the Centers for Disease Control and Prevention (CDC), the Bill and Melinda Gates Foundation (INV-019496, INV-006733), National Institute of Allergy and Infectious Disease of the National Institutes of Health under award number R01AI136664, and the MRC Centre for Global Infectious Disease Analysis (reference MR/R015600/1), jointly funded by the UK Medical Research Council (MRC) and the UK Foreign, Commonwealth & Development Office (FCDO), under the MRC/FCDO Concordat agreement and is also part of the EDCTP2 programme supported by the European Union. The funders had no role in study design, data collection and analysis, decision to publish, or preparation of the manuscript. The findings and conclusions in this manuscript are those of the authors and do not necessarily represent the official position of the funding agencies. For the purpose of open access, the author has applied a Creative Commons Attribution (CC BY) license to any Author Accepted Manuscript version arising.

## Author contributions

M.L.T., J.W.I.-E., L.F.J., and G.M.-R. conceived the study. K.Z., D.L., Z.B.D., S.E.P., and T.S. collated data sources. M.L.T. led the data management, developed the statistical model, and conducted the analysis. D.L., Z.B.D., P.V., S.E.P., K.K., T.S., G.M.-R., and L.F.J. gave expert input on the South Africa VMMC programme and interpretation of data. M.L.T. wrote the first draft of the manuscript with input from J.W.I.-E. All authors revised the manuscript for content and approved the final version for submission.

## Competing interests

The authors declare no competing interests.
