## [Peer Review File · Communications Medicine]

Reviewers' comments:

Reviewer #1 (Remarks to the Author):

This manuscript presents estimates of the probabilities and coverage of male circumcision in South Africa by district, age, and time for the period 2008-2019. The authors developed and applied a Bayesian spatio-temporal model that combined data from five national household surveys (SABSSM surveys in 2002, 2008, 2012 and 2017; and DHS in 2016) and HIV prevention programme data. The main findings were that circumcision coverage has increased from 43% to 64% between 2008 and 2019, and that coverage of medical male circumcision has increased from 19% to 42% over the same time period. Coverage is heterogeneous, varying widely between districts. Only 6 of 52 districts met the target of 80% coverage in 15-49-year olds by 2019. No districts met the target of 90% coverage in 10-29-year olds.

This paper is extremely well written and it was a pleasure to read. This is original and important work that will help inform public health decision making at a national and provincial level. The methods are sound, and it's good to see survey data and programme data being combined to provide important insights like this. I was particularly impressed by the authors deep understanding of the data sources (and the limitations thereof), and their detailed understanding of circumcision (both medical and traditional) in South Africa – this shows up in their careful interpretation of the data and their discussion of the limitations.

I really have no important comments or suggested improvements

Reviewer #2 (Remarks to the Author):

This paper uses Bayesian areal statistical models to derive patterns by age and time of circumcision rates in different regions of South Africa, using both programmatic and survey data. The paper is generally well-written; but there are number of bigger picture things I think should be addressed (and which I've restricted myself to).

MAJOR

Value proposition.

I think the authors should place a greater emphasis on demonstrating the added value of their approach over: 1) using the data, or simple smoothings of data; and 2) previous work. In relation to 1), the plots of survey data in the appendix apparently show coverage at a level that would allow local decision makers to use it directly? Also in relation to 1), the authors jump straight to using a spatial model; it is not clear that this is necessary in the sense of providing an improved fit over simpler (eg non-spatial models) for the additional coding, computational, and interpretational complexity. Lastly in relation to 1), the incorporation of programme data doesn't seem to make a substantial difference. In relation to 2), both times the authors mention previous work (Introduction/line 87, Discussion/line 193) they are a bit vague in how much their use of both data streams advances the state of the art ("often not together", "did not model both data sources formally"). I think a less ambiguous comparison of methods and comparison of results is needed.

Uncertainty.

The main aspect of the results that gives me some pause and I would like better explained is the very low uncertainty, which frankly 'feels' too low (while acknowledging the problem could be with my expectation). This is apparent throughout (from the abstract through to plots of fit to survey data in the appendix). Why are the uncertainty intervals so very much narrower than the CIs of survey data? The figure with/without programme data suggests this is not the whole explanation. Similarly, I imagine there is some meta-analytic sharing of information across ages and times and areal units, but again the uncertainty looks 10x smaller than surveys. In the appendix there is a discussion of the use of survey weights in a pseudolikelihood that should capture the design effects of these surveys - but I wonder if this has been verified? E.g. does the use of this approach for a single survey measure (ie ignoring the modelling of the prevalences) yield estimates for that quantity with comparable uncertainty to that reported? Lastly, some of the results reported are aggregates of the lower level results, and I didn't spot an explanation for how the uncertainties for these aggregate quantities were constructed.

Results.

In my opinion the reporting of results could be improved. I felt table 1 is better placed in the appendix, and figure 6 could be reduced to some numbers in text or on other figures (it also has undefined abbreviations in the caption). While it is methods, figure 8 is duplicated in the appendix and I didn't feel it would really help readers much in understanding and I would suggest removal. I would suggest maintaining consistency of palette meanings between figures (eg not using yellow/green in figure 1 and 2 to mean different things). I also think it's important to have some more presentation in the main paper of the fit to data. At the moment, only figure 7 gives any sense. I think this is important not only because it allows readers to get judge how good the fit is, but also is an opportunity to present the data. Ideally this presentation of fit should also include both survey and programme data, since this is one of the points of the paper. I realise this is a challenge, and but do think it should be possible either through selection or aggregation.

Code availability.

The link to the project GitHub repo doesn't work - perhaps the authors forgot to switch to public - but this does mean I was not able to check it. If licenses permit, and it isn't already in the project repo, it would also be convenient to deposit the collated input data somewhere so that the results can be reproduced.

Responses to Reviewers: Substantial but spatially heterogeneous progress in male circumcision for HIV prevention in South Africa

We would like to thank the editors and reviewers for their useful and constructive feedback. We have addressed each of the points in turn and our responses are below.

Reviewer 1

This manuscript presents estimates of the probabilities and coverage of male circumcision in South Africa by district, age, and time for the period 2008-2019. The authors developed and applied a Bayesian spatio-temporal model that combined data from five national household surveys (SABSSM surveys in 2002, 2008, 2012 and 2017; and DHS in 2016) and HIV prevention programme data. The main findings were that circumcision coverage has increased from 43% to 64% between 2008 and 2019, and that coverage of medical male circumcision has increased from 19% to 42% over the same time period. Coverage is heterogeneous, varying widely between districts. Only 6 of 52 districts met the target of 80% coverage in 15-49-year olds by 2019. No districts met the target of 90% coverage in 10-29-year olds.

This paper is extremely well written and it was a pleasure to read. This is original and important work that will help inform public health decision making at a national and provincial level. The methods are sound, and it's good to see survey data and programme data being combined to provide important insights like this. I was particularly impressed by the authors deep understanding of the data sources (and the limitations thereof), and their detailed understanding of circumcision (both medical and traditional) in South Africa - this shows up in their careful interpretation of the data and their discussion of the limitations. I really have no important comments or suggested improvements.

Response: We thank the reviewer for their time and careful review of our paper. The comments were very kind and we are glad that they see the value in our approach in bringing the two data sources together to aid public health decision making. We particularly appreciated the note that the considerable work in understanding the data sources and the local dynamics in circumcision in South Africa shows through in the reporting; working closely with circumcision programme partners to understand and interpret the data was an important and formative component of this research, which is an important aspect to highlight for potential future applications of our methods.

Reviewer 2

This paper uses Bayesian areal statistical models to derive patterns by age and time of circumcision rates in different regions of South Africa, using both programmatic and survey data. The paper is generally well-written; but there are number of bigger picture things I think should be addressed (and which I've restricted myself to).

Response: We thank the reviewer for carefully reviewing of our paper. The comments and feedback were useful and constructive, and we have found them very helpful. Below we describe the revisions and responses addressing each point.

Value proposition

I think the authors should place a greater emphasis on demonstrating the added value of their approach over: 1) using the data, or simple smoothings of data; and 2) previous work.

In relation to 1), the plots of survey data in the appendix apparently show coverage at a level that would allow local decision makers to use it directly?

Response: We thank the reviewer for the encouragement and helpful suggestions for how to emphasise the added value of our approach. We have revised the Introduction to elaborate on the motivation for approach. We noted that, firstly, household surveys are precise at the national level, but they may be unstable and have large statistical uncertainty at a subnational level (due to sample size) and this will be the case when we start introducing many strata, such as district, age, time and type. This can be seen in the Figures in the Supplementary

Material (Sections C.1-C.3). Secondly, surveys contain a wealth of information on circumcision practice that is not leveraged through typical prevalence modelling approaches, such as age of circumcision and rates by birth cohort over time. Using a time-to-event model allowed us to model local variation in rates of circumcision by age and type, which other approaches do not do. This information will aid decision makers to tailor programmes to the local area, particularly as there is a push from traditional circumcisions to medical circumcisions in traditional male initiation practices. Finally, we would also highlighted that survey data are typically only available every five years so outside of survey years, policymakers rely informally on programme data, which would get re-calibrated upon new survey being published. We integrated both data sources together to produce consistent estimates to aid decision making. These revisions are reflected in the updated the Introduction (Page 4-5, Lines 80–113).

Also in relation to 1), the authors jump straight to using a spatial model; it is not clear that this is necessary in the sense of providing an improved fit over simpler (eg non-spatial models) for the additional coding, computational, and interpretational complexity.

Response: We have revised the Methods Section (Pages 14-15, Lines 419-426) and Supplementary Material (Page 10, paragraph 1) to further elaborate the rationale for our spatial modelling approach. We adopted a spatial modelling approach for several reasons. We sought to pool the age-time information on circumcision and allow areas with little to no data to borrow across regions. As mentioned in the previous response, when considering a large number of strata, survey data are sparse or non-existent in places, so pooling across space-age-time was necessary. We also observed that circumcision practices are spatially correlated (see Figures 5–7 and also Cork *et al.* [1]), with neighbouring regions observing similar male circumcision practices. For example, men in Eastern Cape districts (Buffalo City, Amathole) observe near identical traditional circumcision rates (See Figure S47 in the Supplementary Material).

Lastly in relation to 1), the incorporation of programme data doesn't seem to make a substantial difference.

Response: As also noted in responses above, we considered reconciling the two data sources as a core motivation for our analysis, considering how VMMC programme data are used in practice to monitor programme progress in real time. In South Africa, we noted a general

consistency between both data sources, and we see this as a validation of the modelling approach we have taken. We expect this to have larger importance in potential applications to other countries where there has been large scale-up of VMMC programme implementation in the years since the most recent survey. Furthermore, programmes annually review and update priorities, approaches, and funding from year to year, so using this data in the model provides a more accurate projection in the years following the most recent survey. We have included some more context to this in the Introduction when addressing the above comments (Page 4-5, Lines 80–113).

In relation to 2), both times the authors mention previous work (Introduction/line 87, Discussion/line 193) they are a bit vague in how much their use of both data streams advances the state of the art ("often not together", "did not model both data sources formally"). I think a less ambiguous comparison of methods and comparison of results is needed.

Response: We agree with the reviewer and have provided a more detailed comparison on the approaches by Cork *et al.* [1] and Kripke *et al* [2, 3] and how our approach is different in the Introduction (Pages 4-5, Lines 98-120).

Uncertainty

The main aspect of the results that gives me some pause and I would like better explained is the very low uncertainty, which frankly 'feels' too low (while acknowledging the problem could be with my expectation). This is apparent throughout (from the abstract through to plots of fit to survey data in the appendix). Why are the uncertainty intervals so very much narrower than the CIs of survey data? Similarly, I imagine there is some meta-analytic sharing of information across ages and times and areal units, but again the uncertainty looks 10x smaller than surveys.

Response: We thank the reviewer for this comment; this is an interesting question that we have also considered throughout the development of the model.

Comparing the survey uncertainty ranges compared to posterior coverage uncertainty ranges (which indeed is what we presented in Figures) is not the ideal comparison to assess model fit. We have added analysis of the coverage of posterior predictive distributions, which account

for both the parameter uncertainty and the sampling uncertainty in the observations as a quantitative assessment of model fit to the survey and programme data (See Page 16, Lines 455-486, Page 5, Lines 122-136 and Page 8, Lines 204-230 for details). As part of the checks, we evaluated the coverage the posterior predictive distributions by calculating the proportion of empirical observations that fell within the 50%, 80%, and 95% quantiles of the posterior predictive distribution and found that our model covered 65.4% and 95.0% of the 50% and 95% posterior predictive intervals of medical circumcision coverage survey observations by district and five-year age group (Supplementary Table S5) and 68.6% and 95.6% of traditional circumcision coverage survey observations (Supplementary Table S6).

As the reviewer notes, there are a few reasons why the uncertainty ranges in the modelled estimates are narrower than direct survey estimates at district or national level—some of which are attractive features but others which indicate imperfections of the model. First, as noted by the reviewer, smoothing of circumcision rates between neighbouring districts, over age, and over time (the classic small-area modelling variance shrinkage). We assumed that traditional circumcision rate did not vary over time, which effectively pooled data from the five surveys into a single traditional circumcision rate by age and district. Second, the time-to-event nature of the model more imposes consistency on circumcision reporting by men in the same birth cohort in successive surveys, effectively increasing the sample size within each cohort. For example, men who are surveyed at 22 and 13 in the 2017 and 2008 surveys respectively, are born in 1995 and report a circumcision at age 10 in 2005 have the same trajectory in the model. Third, the VMMC programme data on the number of VMMC conducted in a year relatively precisely identifies the aggregate annual rate of MMC among age 15+ years population (but does not have information about circumcision rates by age).

There are also likely reporting errors or biases in each of the data sources, such as errors in reporting circumcision status, misreporting in the programme data and errors in the population data. These are sources of uncertainty that we are not accounting for currently and we need to think about how these additional uncertainties can be included in the future and we draw attention to these in the discussion (Pages 11-12, Lines 315-341).

We also note that the new Figure 1A illustrates nicely how the magnitude of uncertainty of uncertainty is responsive to the amount of data in a district. For example, there are

some districts (Overberg, Central Karoo, Harry Gwala) where uncertainty is larger due to a small sample size from the household surveys. In districts where uncertainty is lower (Cape Town, Amathole, Johannesburg) there is a very good level of agreement/consistency between surveys as well as the programme data so under our model structure would lead to a low level of uncertainty. See Figures 1 and 7 for more details.

In the appendix there is a discussion of the use of survey weights in a pseudolikelihood that should capture the design effects of these surveys - but I wonder if this has been verified? E.g. does the use of this approach for a single survey measure (ie ignoring the modelling of the prevalences) yield estimates for that quantity with comparable uncertainty to that reported?

Response: The pseudo-likelihood approach of replacing observed counts by survey-weighted counts scaled by an effective sample size is commonly used in literature modelling complex survey data (for example both Cork *et al.* [1]). This is an active area of research, see for example Gelman [4] highlighting the difficulties of accounting for a complex survey design in Bayesian models and Chen, Wakefield, and Lumley [5] comparing alternative approaches. While the approach has generally provided satisfactory results in our and other's experience, we agree that this is an important area for further research and have highlighted this in the Discussion (Pages 11, Lines 330–333).

Lastly, some of the results reported are aggregates of the lower level results, and I didn't spot an explanation for how the uncertainties for these aggregate quantities were constructed.

Response: The posterior aggregates were obtained from taking joint samples from the posterior distribution of circumcision coverage in each region-age-time-type stratum. Each sample was aggregated into the quantity of interest (e.g. population weighted circumcision coverage in men aged 15-49) resulting in a samples from the posterior distribution of the quantity of interest. For each output, the posterior mean, median, standard deviation, and quantile-based 95% credible intervals (CI) were computed from the corresponding posterior distribution. We have clarified this in the paper (Pages 16–17, lines 476–488) and the Supplementary Material (Page 14, paragraph 4).

Results

In my opinion the reporting of results could be improved.

I felt table 1 is better placed in the appendix.

Response: We understand that this table contains a considerable amount of information and detail. However, we feel that the results presented are important to communicate the detail of results that are available from the model to a wider readership, and, importantly, present key information for the important audience with interest in how circumcision coverage has changed in South Africa and would like to use these results to inform national programming. As such, we have chosen to retain this table in the main text. We are, however, open to further guidance on this point from the Editor.

Figure 6 could be reduced to some numbers in text or on other figures (it also has undefined abbreviations in the caption).

Response: Figure 6 (now Figure 8) reported the distribution in age at circumcision across provinces and by circumcision type. Estimating the distribution and spatial variation in age at circumcision is an important and novel output from our model, which alternative models are unable to capture [1–3]. We placed a larger emphasis on this in the Introduction in response to the other comments above, and think that this figure is of particular importance to the readership of Nature Communications Medicine in highlighting the heterogeneity of circumcision practice in South Africa. We are however open to doing as the reviewer suggests and leave this up to the discretion of the Editor. We have also removed the abbreviations MMC-nT and TMIC from the figure caption.

While it is methods, figure 8 is duplicated in the appendix and I didn't feel it would really help readers much in understanding and I would suggest removal.

Response: As suggested by the reviewer, we have removed the model structure diagram (previously Figure 8) from the main text of the paper, retaining it only in the Supplementary Material.

I would suggest maintaining consistency of palette meanings between figures (eg not using yellow/green in figure 1 and 2 to mean different things).

Response: We thank the reviewer for highlighting this and have changed the colour schemes of the figures to ensure consistency when showing results by circumcision type, time, coverage, age group and model.

I also think it's important to have some more presentation in the main paper of the fit to data. At the moment, only figure 7 gives any sense. I think this is important not only because it allows readers to get judge how good the fit is, but also is an opportunity to present the data. Ideally this presentation of fit should also include both survey and programme data, since this is one of the points of the paper. I realise this is a challenge, and but do think it should be possible either through selection or aggregation.

Response: As suggested by the reviewer, we have increased the content and focus on fit to data in the main paper through three revisions: Firstly, we have updated the previous Figure 7, now Figure 1, to illustrate the model fit to household survey data by space, age, time and type. Figure 1A now displays the effects of adding the programme data on a district level over time, Figure 1B shows the model fit by the circumcision type on a province level and Figure 1C shows the model fit in 5 year age groups over time at the national level. We have also added a new Figure 2 displaying the number of MMCs predicted in the models with and without survey data along with the number of reported MMCs from VMMC programmes to demonstrate the effect of adding programme into the model. Finally, we have added posterior predictive checks of our models to the paper. We have revised the methods section (Page 16, Lines 455–466) and the Results section (Page 5, Lines 122–136 and Page 8, Lines 204–230) accordingly.

Code availability

The link to the project GitHub repo doesn't work - perhaps the authors forgot to switch to public - but this does mean I was not able to check it.

We thank the reviewer for highlighting this and apologise for the oversight. We have made created a new public Github repository that contains the code for the model and analysis in this paper (<https://github.com/mrc-ide/zaf-circumcision-paper>) and archived the code with Zenodo and is available from <https://doi.org/10.5281/zenodo.8160045>. This new repository

is linked in the Code Availability Statement.

If licenses permit, and it isn't already in the project repo, it would also be convenient to deposit the collated input data somewhere so that the results can be reproduced.

Unfortunately, we are not able to make the full input data publicly accessible due to terms of data use agreement with the South Africa National Department of Health and the Human Sciences Research Council. In the Data Access Statement, we have provided information and links to register or request each of the data sources used in the analysis. However, the R package noted in the Code Availability Statement (<https://github.com/mrc-ide/threemc>), implementing an extended version of this model, will include example data sets capable of demonstrating the model.

References

1. Cork, M. A. *et al.* Mapping male circumcision for HIV prevention efforts in sub-Saharan Africa. *BMC medicine* **18**, 1–15 (2020).
2. Kripke, K. *et al.* Age Targeting of Voluntary Medical Male Circumcision Programs Using the Decision Makers' Program Planning Toolkit (DMPPT) 2.0. *PloS one* **11**, e0156909 (2016).
3. Kripke, K. *et al.* Cost and Impact of Voluntary Medical Male Circumcision in South Africa: Focusing the Program on Specific Age Groups and Provinces. *PLoS One* **11**, e0157071 (2016).
4. Gelman, A. Struggles with survey weighting and regression modeling (2007).
5. Chen, C., Wakefield, J. & Lumely, T. The use of sampling weights in Bayesian hierarchical models for small area estimation. *Spatial and Spatio-temporal Epidemiology* **11**, 33–43 (2014).

REVIEWERS' COMMENTS:

Reviewer #1 (Remarks to the Author):

Considered responses to reviewer 2 and appropriate revisions to manuscript. No further comments.

Reviewer #2 (Remarks to the Author):

Thanks to the authors - I'm happy with these responses & changes.

One minor note: there are various references that contain errors (eg "Organization, W.H") or are incomplete (e.g. Gelman 2007); but I'm sure these will be sorted out in due course.